# An Unsupervised Neural Network Feature Selection and 1D Convolution Neural Network Classification for Screening of Parkinsonism

**DOI:** 10.3390/diagnostics12081796

**Published:** 2022-07-25

**Authors:** Tariq Saeed Mian

**Affiliations:** Department of IS, College of Computer Science and Engineering, Taibah University, Madinah Al Munawara 43353, Saudi Arabia; tmian@taibahu.edu.sa; Tel.: +966-563304330

**Keywords:** Parkinson’s disease, ML, linear discriminate analysis, dimensionality reduction, principal component analysis, neural network, random forest, support vector machine, logistic regression

## Abstract

Parkinson’s disease (PD) is the second most common neurodegenerative disorder after Alzheimer’s disease. It has a slow progressing neurodegenerative disorder rate. PD patients have multiple motor and non-motor symptoms, including vocal impairment, which is one of the main symptoms. The identification of PD based on vocal disorders is at the forefront of research. In this paper, an experimental study is performed on an open source Kaggle PD speech dataset and novel comparative techniques were employed to identify PD. We proposed an unsupervised autoencoder feature selection technique, and passed the compressed features to supervised machine-learning (ML) algorithms. We also investigated the state-of-the-art deep learning 1D convolutional neural network (CNN-1D) for PD classification. In this study, the proposed algorithms are support vector machine, logistic regression, random forest, naïve Bayes, and CNN-1D. The classifier performance is evaluated in terms of accuracy score, precision, recall, and F1 score measure. The proposed 1D-CNN model shows the highest result of 0.927%, and logistic regression shows 0.922% on the benchmark dataset in terms of F1 measure. The major contribution of the proposed approach is that unsupervised neural network feature selection has not previously been investigated in Parkinson’s detection. Clinicians can use these techniques to analyze the symptoms presented by patients and, based on the results of the above algorithms, can diagnose the disease at an early stage, which will allow for improved future treatment and care.

## 1. Introduction

Parkinson’s disease (PD) is a slow progressing neurodegenerative disorder, causing impaired motor function with slow movements, tremors, and gait and balance disturbances. Various non-motor symptoms are common, and include disturbed autonomic function with orthostatic hypotension, constipation and urinary disturbances, sleep disorders, and neuropsychiatric symptoms. The onset of PD is insidious, with peak age of onset being 55–65 years. The disease has drastic effects on the lives of millions of people all over the world [1]. The progression rate of PD is slow; however, as it progresses, the affected person loses control over their movement, resulting in serious issues. The main cause is the loss of dopaminergic neurons in the substantia nigra and the decrease in the level of dopamine in the striatum. PD clinical diagnosis is possible from the following four cardinal motor symptoms: a tremor at rest, bradykinesia, postural instability, and rigidity. The diagnosis of PD based on symptoms becomes possible when almost 60% of dopaminergic neurons are already dead [2].

PD symptoms can vary from person to person due to its variety of symptoms, as exemplified by the multiple types of possible PD tremors, such as limb rigidity, tremors in the hands, and balance and gait problems. PD symptoms are generally classified into two categories: motor (movement-related) and non-motor (unrelated to movement). Patients experiencing non-motor symptoms face more severe conditions compared with those with motor symptoms. PD is ranked as the 14th leading cause of death in the United States according to the report of Central Disease Control and Prevention (CDC) [3], and is the second most common neurological disease after Alzheimer’s disease [4]. There is a dire need for a mechanism to detect and diagnose PD at an early stage, as the timely detection of PD facilitates rapid treatment and removes or lessens PD symptoms. Early-stage PD detection is a key element to slowing down its progression, and increasing the success rate of patient recovery.

However, researchers are still attempting to identify the most appropriate treatment for Parkinson’s disease [5]. Multiple diagnostic tests and various symptoms are used in combination with PD detection. The biomarkers are investigated for early detection to slow down the disease process. Different approaches were developed based on voice data for PD detection [6,7,8], force tracking data [9], and gait patterns. The current methods for evaluating Parkinson’s disease mostly rely on human efforts and expertise [10]. In the PD diagnosis literature, there is an immense body of work from researchers using speech measurement for general voice disorders and in PD [11,12,13]. In these studies, speech tests are recorded using a microphone, and then the recorded signals are analyzed using software algorithms to detect the properties of these signals.

As the median age of the population is increasing, the number of elderly people is rising. This scenario is contributing to the number of people affected by neurological disorders, and there is a growing need for fast and effective measures to monitor them. It is reported that nearly 90% of PD sufferers experience impairment of the vocal cords, which causes dysphonia; i.e., having an abnormal voice. Articulatory deficits can also manifest along with vocal impairment in the more severe stage [14]. Voice disorder detection at an early stage can play an important role in predicting the progression of the disease. The traditional sound measurement standards are shimmer, jitter, noise to harmonics ratio, pitch, and pressure level of absolute sound. Other measurements were also proposed that are inspired by non-linear dynamical systems theory. The tools recurrence density entropy (RPDE) and detrended fluctuation analysis are used for the detection of voice disorders [15].

The convolutional neural network (CNN) can extract features automatically and brought revolution, especially in the field of computer vision. The common applications of CNN in computer vision tasks are [16] image classification [17], object detection [18], image segmentation [19], medical domain [20], agriculture domain [21], scene understanding [22], and text classification.

This study aims to use the unsupervised autoencoder feature selection technique, and then pass the optimal features to supervised ML models. We also use a 1D convolutional neural network model to verify the deep learning solution for PD detection. The proposed approach, convolutional neural network-1D (CNN-1D), has better results when compared with the autoencoder feature selection technique.

### 1.1. Motivation

PD is a slow progressive neurodegenerative disease that affects almost 2% of people over the age of 65, and is the second most commonly found neurodegenerative disease after Alzheimer’s disease. PD affects almost 6.3 million people all over the world. PD patients lose their dopamine-producing neurons, which causes motor and non-motor symptoms. PD is characterized by motor and non-motor symptoms, and sufferers can exhibit the following; movement slowness, sidedness, tremors, vocal disorders, and jerkiness. There is still no effective cure and no means of slowing the progression of this fatal disease, and early detection is key to treatment.

The hypothesis of this research study was that PD could be detected in its early stages in a person by feeling the changes in the characteristics of speech as they talk. Such changes could be used to distinguish and classify people with PD disease and without PD disease.

### 1.2. Research Problem and Objective

Due to a rise in PD sufferers, the early detection of PD using an automated approach is becoming increasingly vital to improve the level of care for PD sufferers. The voice-based dataset contains important features for analysing PD. These type of data undergo many processes such as scaling, normalization, aggregation, and sampling to obtain an accurate prediction in the health care system. Previous researchers conducted a large number of studies to address and propose a solution for PD detection. The deep-learning-based solution brought about revolutions, and offers new opportunities for researchers to implement novel techniques to readily detect PD at an early stage. This study compares the performance of different deep-learning and machine-learning techniques. The aim of this paper is to address the following research questions:

**RQ1:** Are the autoencoder feature selection techniques providing better results compared to traditional ML algorithms [23]?

**RQ2:** Is the proposed dimensionality reduction technique providing better results than the recursive feature selection technique [24]?

**RQ3:** Is the proposed unsupervised autoencoder feature selection providing better results than the 1D-convolution neural network that has automatic feature selection?

The major contributions of this study are as follows:The proposed CNN-1D is the first approach used for the classification of PD, and proves that the deep-learning model returns outstanding results in a large amount of data without feature selection;We conducted a comparison between a statistical recursive feature elimination technique (RFE) and a deep-learning-based unsupervised autoencoder to select the optimal subset of features;Our proposed approach performs analysis between different dimensionality reduction techniques and provides outstanding results.

The remaining portion of the paper is organized as follows. Section 2 reviews the literature, detailing previous studies of this domain. In Section 3, a brief description of the dataset, proposed methodology, and the proposed method can be found. Section 4 describes the results of this work and, finally, Section 5 describes the conclusion of the study and future direction.

## 2. Literature Review

PD is a fatal neurodegenerative disease, which is still impossible to diagnose in its early stages, where it could easily be treated, and the degeneration process could be slowed down. PD affects the quality of sufferers’ lives all over the world [25]. PD symptoms vary from individual to individual, as it has a variety of symptoms. Some patients experience tremors mainly at rest, such as tremors in their hands, irregular sequence of foot movement, and issues with balancing their body. Two types of symptoms are common in PD: motor and non-motor. Patients with non-motor symptoms are more affected than patients with motor symptoms. Non-motor symptoms include sleep disorders, depression, cognitive impairment, and loss of sense of smell. It is reported in the clinical literature that one of the most common symptoms of PD is vocal cord impairment, resulting in vocal loudness, frequency abnormality, and instability. Vocal impairments and voice breaks are the key factors that provide information about PD patients. Speech processing techniques are used to find the anomalies to detect PD. In 2015, the Movement Disorder Society published their research on a clinical diagnostic criterion for PD [26]. This criterion proved helpful in excluding the possibility of any other diseases that may have a similar set of symptoms, and requires the patient to have a minimum of two clinical symptoms from the given set of symptoms for a positive diagnosis of PD. The neurologist assessment for PD is a subjective assessment based on patient interview and some physical tests. The aforementioned criterion does not have a standard format of test sequence and questionnaire. In 1980, the Unified PD Rating Scale (UPDRS) was developed, which was later modified in 2008 [27]. Research studies extensively use UPDRS-based symptoms for categorizing PD [28,29].

Feature selection is a technique to identify the features best suited for the model from a large set of possible features. Feature selection reduces the computational requirements and enhances the generalization of the ML method. Three feature selection methods, (i) sequential forward feature selection (SFS), (ii) sequential backward feature selection (SBS) [30], and (iii) minimum redundancy maximum relevance (mRMR) [31], are used in this research, and their performance is compared. The maximum relevance feature selection technique selects features based on maximum mutual information by adding additional features in the subset. However, the selected subset may contain redundant data. The minimum redundancy maximum relevancy (mRMR) optimizes the relevance and reduces the redundancy of the selected features. In the mRMR feature selection technique, two features are compared based on mutual information, and the one that affects the result the most (maximum relevance) is selected, yet they have a different minimum redundancy. The mRMR has been used extensively in the classification and detection of early PD from voice recordings [32], classification of pedestrian information, and identifying physical activities [33]. The mRMR could also be suitable for assessing gait features of PD patients.

The past decade witnessed a phenomenal growth in the literature in the field application of ML in medical diagnosis, such as the diagnosis of PD, using different features such as vocal impairment. The following table lists some of the studies where ML was used to diagnose PD. Table 1 provides a summary review of the literature.

## 3. Methodology

The proposed methodology is based on the following main steps: data pre-processing, dimensionality reduction, classifiers training on training data, and classifiers evaluation on test data. To the best of our knowledge, the research scope of the proposed study makes it the first study to use a non-linear dimensionality reduction technique autoencoder in Parkinson’s disease detection. The autoencoder neatly compressed the features and sends striking performance results to the ML classifiers. We explored the proposed technique with reference to the linear dimensionality reduction techniques, and conclude that the proposed technique provides outstanding results in term of accuracy and F1 measure.

The overall proposed methodology can be illustrated as:

### 3.1. Data Set

In this research, a vocal-based dataset was used that contains healthy and affected individuals’ vocal recording instances. The dataset for this study was taken from California University, Irvine ML dataset library. This dataset has records for 81 female and 107 male PD patients (i.e., 188 patients altogether). The healthy group has 64 samples (23 male and 41 female), with an age range of 41 to 82 years old. The final version of the dataset contains 754 attributes with 756 records [44,45]. Table 2 provides dataset description.

We also analyzed the class distribution of the dataset, which was skewed and means our dataset was imbalanced. The classification accuracy tends towards the majority class. To address the issue of the imbalanced dataset, we used an oversampling technique, in which class distribution 0 and 1 are equal. Oversampling increases the distribution in the minority class and makes an equal distribution of both classes. Figure 1 shows the class distribution of the target value without oversampling.

The dataset has a split ratio of 30:70, where 30% is the test data while 70% is the training. Training data is used for models training, while test data is used for model evaluation. We used anaconda distribution environment (Jupyter) for the implementation of the proposed models and pre-processing.

### 3.2. Proposed Models

In this study, two different techniques for Parkinson’s disease detection at an early stage were used. We used comparative analysis of the unsupervised neural network feature selection autoencoder that compressed the input features, and then passed them to supervised ML algorithms for classification, as ML algorithms are not suitable for automatic feature engineering, while deep-learning algorithms are capable of automatic feature selection. We also used Conv1D for the early detection of PD.

#### 3.2.1. Autoencoder

An autoencoder is an unsupervised dimensionality reduction neural network technique, where the neural network is powered to learn the representations. An autoencoder has two main parts: the encoder and the decoder. The encoder encodes the input features into a lower dimensional space, called latent vector, and the decoder decodes the latent vector into its original form. The loss function is used to minimize the difference between input features and output features, (see Figure 2).

In this research, a sparse autoencoder was used with a single hidden layer representation, Table 3 lists autoencoder parameters. The input features were encoded under this single hidden layer, and effort was made to minimize the error and generate a flattened representation with optimal score [39]. The sparse autoencoder is an autoencoder model with the addition of a sparse penalty term and adds more constraints for the extraction of the latent vector and enhanced feature learning. In the network, the activation function sigmoid was used. In order to reduce the activation functions value in the hidden layer neuron, a sparse penalty was embedded [40]. Due to this sparsity, some undesired activation in the hidden layer is possible. The activation function can be represented as:(1)$$a=sig\left(Wx+b\right)$$
where:

*W* = weight matrix.

*b* = deviation vector.

The neuron activation function in the hidden layer can be represented as follows:(2)$$p_{j=}\frac{1}{n}\overset{n}{\underset{i=1}{\sum }}\left[a_{j}x\left(i\right)\right]$$

(KL) divergence method is used as a penalty term. KL divergence can be represented as follows:(3)$$kl(p||p_{j})=\mathrm{pln}\frac{p}{p_{j}}+\left(1-p\right)ln\frac{1-p}{1-rho_{j}}$$

The deviation causes a gradual increase in the value of Kullback–Leibler (KL), and it becomes zero as it deviates from *p*. The loss value of this network is represented by C(w, b). The following equation shows the loss function with the addition of the sparse penalty term:(4)$$C_{\mathrm{sparse}\;=\;C\left(w,\;b\right)+\beta \overset{S_{2}}{\underset{j=1}{\sum }}\mathrm{kl}(p|{|p}_{j})}$$
where β is the weight of the sparse penalty term and S_2_ is the neuron count in the internal layer [38].

#### 3.2.2. Logistic Regression Model

Logistic regression is a ML algorithm that can only solve the classification tasks using the probability value 0.5 to predict the target variable. It uses the complex cost function, known as sigmoid function or logistic function. Logistic regression analysis has become a powerful predictor in health care research over the last few decades [47]. It is mostly used where the occurrence of a binary outcome is predicted for one or more variables [48]. The useful property of logistic regression is that its output always lies between 0 and 1. The basis of a logistic regression is the sigmoid function [49] that is represented as:(5)$$f\left(x\right)=\frac{e^{x}}{1+e^{x}}=\frac{1}{1+e^{-z}}$$
where:

*e* = Euler number, 

*x* =Value of independent variable *X*.

#### 3.2.3. Support Vector Machine (SVM)

This is a supervised ML technique that is widely employed for classification, pattern recognition, and regression analysis. In SVM, the training data are treated as data points in space, where different classes are divided by a hyper-plane that is far from the nearest data point. New input features are treated as training points, and classified according to their category as they fall into the hyper-plane. In cases where the data points are not linearly separable, kernel tricks with different possible kernel functions, such as polynomial or radial basis functions, are used. These kernels are helpful to map the high dimensionality feature space, and determine a possible high dimensional hyper-plane. The advantage of the support vector machine is that it is effective for high dimensional spaces in which data points are divided into subsets of training points by creating a decision boundary. SVM is versatile because of various possible kernel functions. SVM has the demerit that it does not have probability estimates for classification problems, and regularization terms are not necessary to avoid over-fitting. The SVM model diagram shows the margin between predictor and target variables.

#### 3.2.4. Naïve Bayes Model (NBM)

This model is a supervised ML approach that can solve classification tasks. The naïve Bayes model works based on conditional probability, by using learning probability of certain features belonging to a particular group. The naïve Bayes model is also known as “Naïve” as it assumes that the occurrence of a particular feature is independent of another feature occurrence. The naïve Bayes theorem finds the probability of event occurrences that have already occurred. The naïve Bayes model maps a series of input features *X* = [*x*1, *x*2, ..., *x*n] to a series of probability value of *Y* = [*y*1, *y*2, *y*3, ..., *y*n]. For a particular set of observations *X* = (*x*1, *x*2, ..., *x*n), we determined those in which *Y* belongs to yi, and made the classification of the particular object. We must find the highest value of yi.

The mathematical representation of a naïve Bayes theory is given below:(6)$$P\left(A|B\right)=\frac{P\left(A\right)\;*\;P\left(B|A\right)}{P\left(B\right)}$$

#### 3.2.5. Random Forest Model (RFM)

This model is an ensemble technique that can solve classification, as well as regression, tasks. RFM uses a modified tree-learning algorithm during the learning process, selecting a random subset of the features. The technique only determines a random subset of variables to find the best split at each node. The input vector is fed to each tree in the RFM for the classification task, and each tree votes for a class. The RFM selects the class that has the highest number of votes. The RFM can handle large input datasets compared to other models. The random forest model overcomes the problem of over-fitting by averaging or combining the results of different decision trees. RFM has the same hypermeters as in the decision tree model or bagging models.

#### 3.2.6. Convolutional Neural Network-1D

Convolutional neural network comes under the umbrella of deep learning. Deep learning is a subfield of ML, and the algorithms of deep learning have the automatic feature selection and good performance over a large dataset. The deep-learning model performs well in speech, language translation, and object detection. Conv2D is used for 2D data such as images and video. CNN-1D is used for 1D and sequential type datasets. In the proposed study, two layers of Conv1D with batch normalization are used. Each layer also has the dropout function to overcome the over-fitting. Layers 1 and 2 also have a 1D max-pooling layer that contains only high feature values. The third is a fully connected layer and the fourth is the output layer. Sigmoid activation function was used because the target attribute is a binary classification task. The number of iterations is 20, with a learning rate of 0.0001. The CNN-1D architecture is given below in Figure 3.

## 4. Results and Discussion

### 4.1. Evaluation Metrics

To verify the performance of the proposed algorithms, we used the following classification metrics:

**Accuracy:** It is the simple ratio between the numbers of correctly classified samples out of total samples. The formula of accuracy score is:(7)$$\mathrm{Accuracy}:\frac{\mathrm{TN}+\mathrm{TP}}{\mathrm{TN}+\mathrm{FP}+\mathrm{FN}+\mathrm{TP}}\times 100\%$$

**Precision:** It is the fraction of the correctly classified samples from the total classified samples. The precision formula is:(8)$$\mathrm{Precision}:\;\frac{\mathrm{TP}}{\mathrm{TP}+\mathrm{FP}}\times 100\%$$

**Recall:** It is the fraction of the correctly classified samples from the total classified samples. The formula of recall is:(9)$$\mathrm{Recall}:\;\frac{\mathrm{TP}}{\mathrm{TP}+\mathrm{FN}}\times 100\%$$

**F1 Score:** It is the harmonic mean of precision and recall. It can be expressed by the following formula:(10)$$F1-\mathrm{Score}=\frac{2\times \mathrm{precision}\times \mathrm{recall}}{\mathrm{precision}+\mathrm{recall}}$$

### 4.2. Discussion

**RQ1:** Are the autoencoder feature selection techniques providing better results compared to traditional ML algorithms [23]?

To verify the proposed technique, we made comparisons with traditional ML classifiers with dimensionality reduction (autoencoder) and without dimensionality reduction techniques. The experimental study proves that logistic regression and support vector machine models with the dimensionality reduction technique provide better results in terms of accuracy score, (see Table 4).

**RQ2:** Does the proposed dimensionality reduction technique provide better results than the recursive feature selection technique [24]?

The basic aim of dimensionality reduction and feature selection is to reduce the number of features from the original set of features. The feature selection is the subset of the original dataset, while dimensionality reduction transforms the features into lower dimensions. Dimensionality reduction and feature selection have various techniques. In this study, we evaluate the comparisons of the autoencoder with the study [50] that used the recursive feature elimination technique. We found that the proposed model with dimensionality reduction provides better results compared to the feature selection technique RFE, see Table 5.

**RQ3:** Does the proposed unsupervised autoencoder feature selection provide better results than the 1D-convolution neural network that has automatic feature selection?

In this paper, we also investigate the power of the deep-learning model compared with dimensionality reduction ML models. In our experimental study, we conclude that the 1D-CNN model shows good performance compared to manual dimensionality reduction of supervised ML models (see Table 6).

We examined the comparisons between linear dimensionality reduction techniques (PCA and LDA) with non-linear dimensionality reduction techniques (autoencoder). We found that the autoencoder provides the optimal features to the ML model, and obtains higher results in terms of accuracy and F1 measure. The overall comparison of dimensionality reduction is given below in Table 7.

The results of the autoencoder with supervised ML are analyzed in an ROC curve. The ROC curve analysis shows (Figure 4) that the random forest model has the highest AUC value, 0.929%.

The results are analyzed in terms of a confusion matrix (Figure 5, Figure 6, Figure 7, Figure 8 and Figure 9). The confusion matrix provides information about the correctly classified and misclassified number of instances.

Caution should be applied in concluding that our approach is perfect for PD detection at early stage, as this study was conducted using only a single dataset. We also did not analyze the various data balancing approaches to avoid misclassification of the results. The classification accuracy could be further improved in future studies by using different hyper-parameter techniques.

## 5. Conclusions

Initial stage of PD detection is a very important step in gaining a better understanding of the causes of the disease, thus, ensuring the correct measures are implemented and therapeutic interventions are initiated. This study is an effort to outline a broad review of the diagnosis system of PD, by applying various robust ML techniques. The different ML dimensionality reduction techniques are evaluated to achieve optimal features, and comparisons are made to obtain the best performance on the benchmark dataset. In this study, we performed analysis of PD in four different ways: (a) evaluation of PD detection without any feature selection technique; (b) analysis of four different machine-learning-based methods on three different dimensional reduction techniques; (c) we also performed analysis of recursive feature elimination technique with dimensionality reduction techniques; and (d) in last section of our experimental approach, we also conducted our evaluation on 1D-CNN with traditional machine-learning models. We conclude that the 1D-CNN model shows better results compared to the manual feature selection technique; the 1D-CNN model shows the highest accuracy of 0.85%, and logistic regression with autoencoder dimensionality reduction shows the highest accuracy of 0.75%. Based on this experimental evaluation, we can say that deep-learning-based 1D-CNN shows better performance as compared to manual-feature-selection-techniques-based machine-learning models. The reason behind this is that the deep-learning model performs better with a large number of features, and has the capability of an automatic feature selection. Our technique has some limitations: we conducted analysis on only one dataset, and we do not consider the dataset balancing methods. In future studies, we are interested in exploring the ensemble and boosting techniques with different feature selection techniques on more than one PD dataset. The outcome of this work can be viewed as a promising first step towards the application of cutting-edge research for the early detection of PD.

## Figures and Tables

**Figure 1 diagnostics-12-01796-f001:**
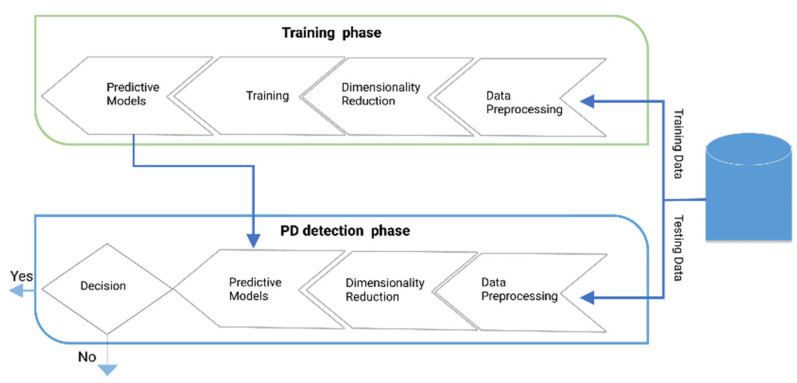
Flow chart of PD detection.

**Figure 2 diagnostics-12-01796-f002:**
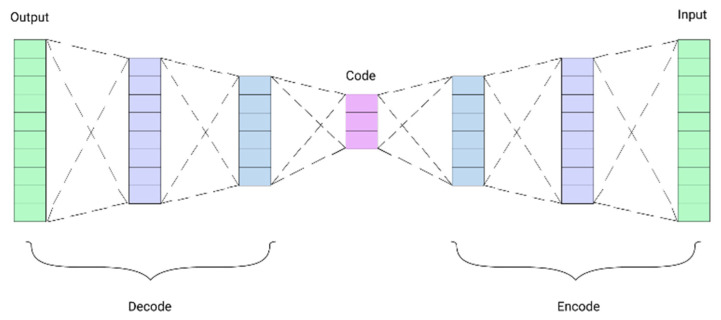
Structural diagram of autoencoder model, adopted from [46].

**Figure 3 diagnostics-12-01796-f003:**
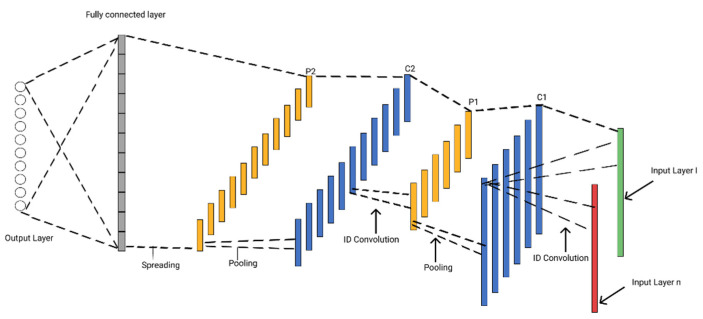
1D-CNN architecture.

**Figure 4 diagnostics-12-01796-f004:**
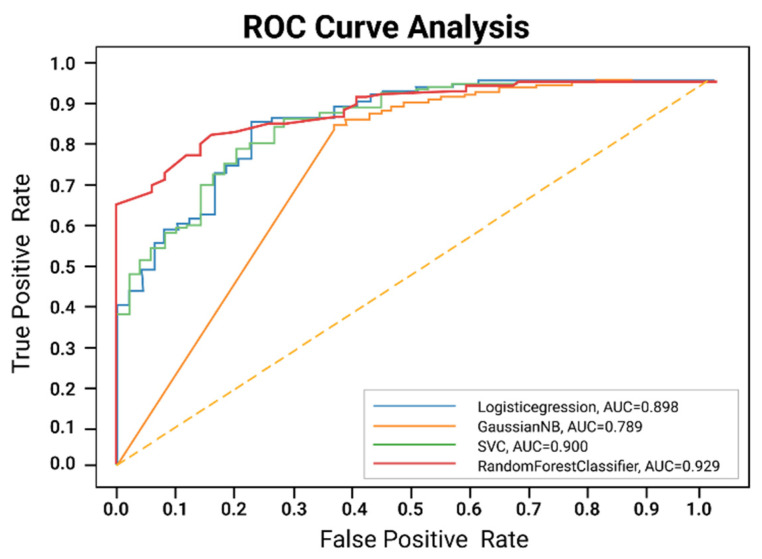
Proposed model with autoencoder ROC curve.

**Figure 5 diagnostics-12-01796-f005:**
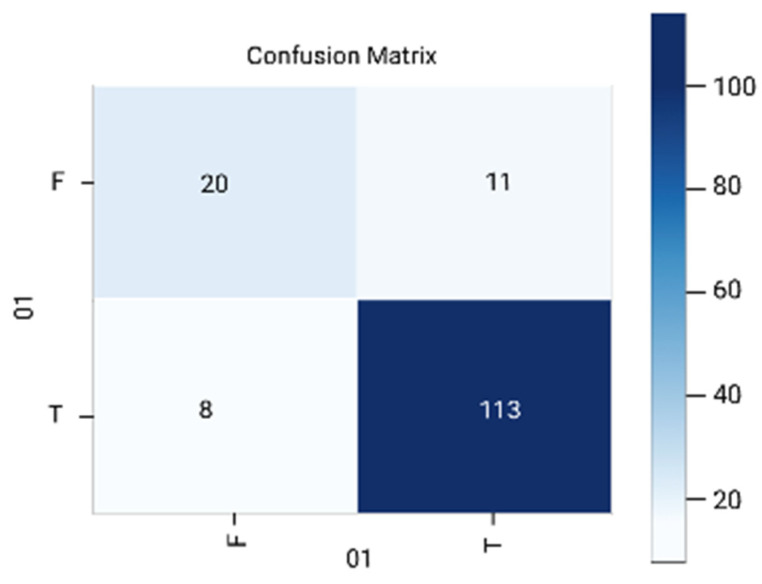
Logistic regression.

**Figure 6 diagnostics-12-01796-f006:**
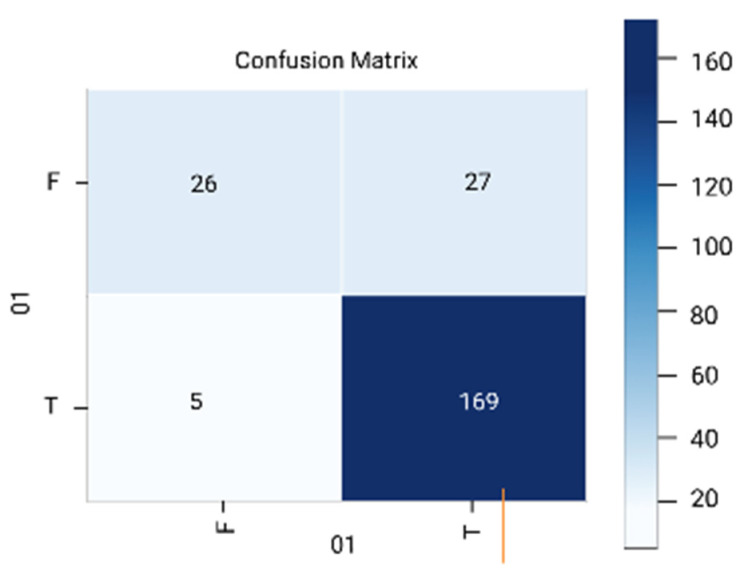
Support vector machine.

**Figure 7 diagnostics-12-01796-f007:**
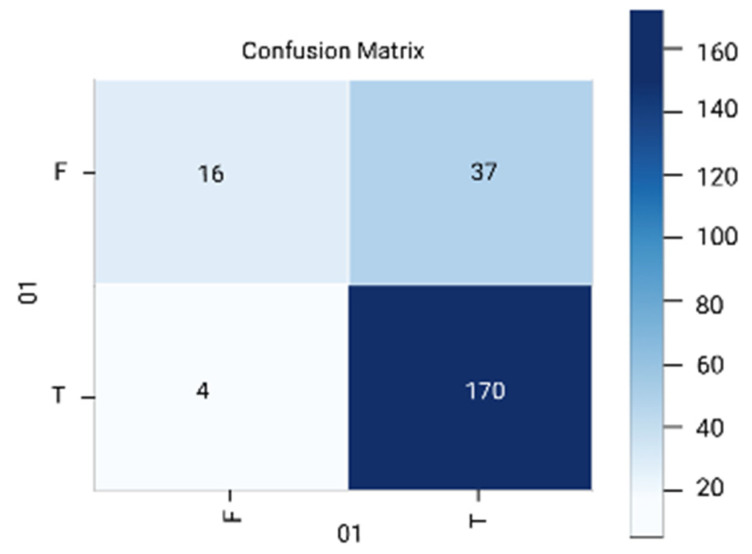
Random forest.

**Figure 8 diagnostics-12-01796-f008:**
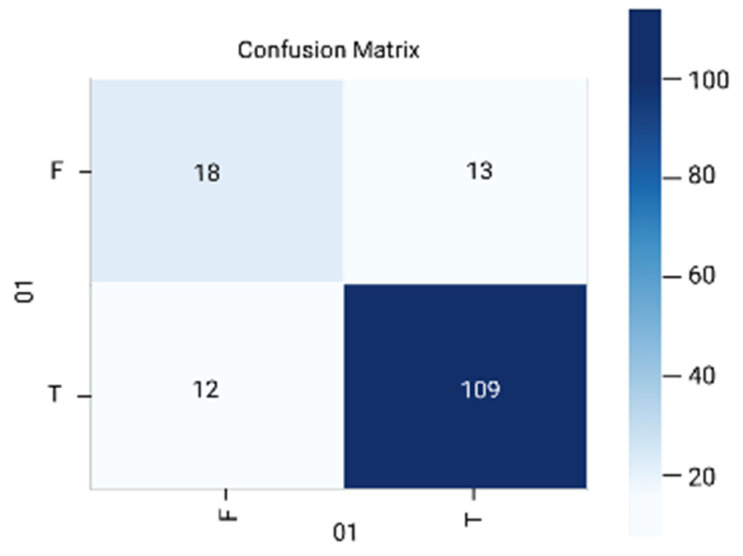
Naïve Bayes.

**Figure 9 diagnostics-12-01796-f009:**
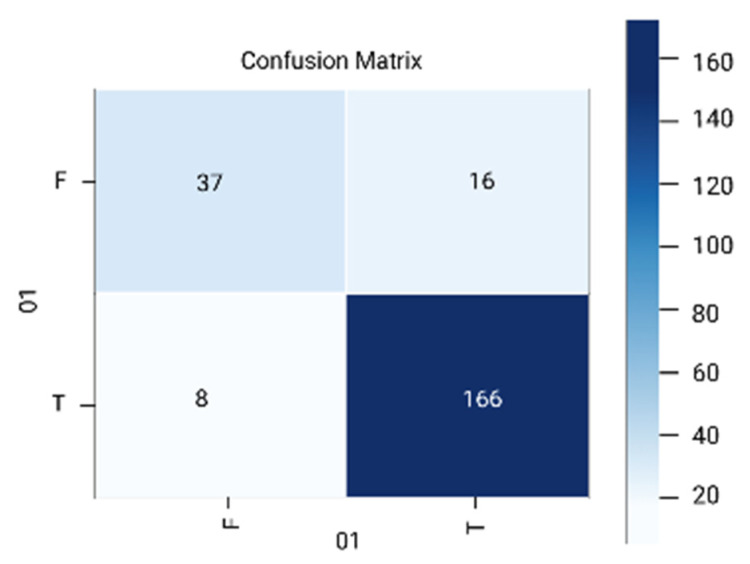
1D-CNN.

**Table 1 diagnostics-12-01796-t001:** Review of the literature.

Study	Techniques Used	Remarks
Tsanas and Athanasios, et al. [34]	Relief and local learning-based feature selection (LLBFS), minimum redundancy maximum relevance (mRMR), and least absolute shrinkage and selection operator	Features such as HNR, shimmer, and expiation of vocal fold produces 98.6% precision rate, while LLBFS produces the feature set with the lowest classification error.
Rouzbahani et al. [35]	Fisher’s discriminate ratio, correlation rates, and *t*-test.	Highest accuracy rate of 94% observed in kNN.
Parisi et al. [36]	Multi-layer perceptron (MLP) with custom cost function and Lagrangian support vector machine (LSVM) are used for classification.	The proposed algorithm achieves 100% of accuracy rate.
Abdullah et al. [37]	DNN classifier with softmax layer and stacked-auto-encoder (SAE).	Several different datasets were used to test the proposed model. DNN classifier is found to be the most suitable classifier for the early diagnosis of the disease.
Timothy et al. [38]	DNN classifier. Minimum redundancy maximum relevance and MFCC used to extract the features.	The study reports 85% accuracy for DNN model.
Mathur et al. [39]	kNN, Adaboost.	The study reports 91.28% classification accuracy for kNN and Adaboost
Yasar et al. [40]	Artificial neural networks (ANN).	The study reports that the proposed model achieves 94.93% accuracy in identifying diseased individuals.
ShuLih et al. [41]	Thirteen-layer CNN architecture was used on a dataset of EEG signals (20 normal subjects and 20 PD sufferers).	88.25% accuracy reported for the proposed CNN architecture.
Laura et al. [42]	Logistic regression model on smell identification and Sniffin’ Sticks test.	Model shows 82.8% accuracy for smell identification and 85.3% accuracy for Sniffin’ Sticks test.
Daryl et al. [43]	SVM and random forest algorithm.	SVM shows AUC of 92.3% and an accuracy of 85.3%. A random forest achieves 76.3% AUC with 75.6% accuracy.

**Table 2 diagnostics-12-01796-t002:** Dataset description.

Detail	Source Information
Dataset property	UCI ML Repository
Dataset name	PD
Dataset attributes	754
Dataset records	756
Target variable	(0: control, 1: PD). Binary class problem
Task	Binary classification

**Table 3 diagnostics-12-01796-t003:** Sparse autoencoder parameters.

Parameters	Values
No. of epochs	200
Weight decay	20^−5^
Optimization technique	Lbfgs
Sparse penalty weight	3
Sparsity	0.1

**Table 4 diagnostics-12-01796-t004:** Classifiers with dimensionality reduction and without dimensionality reduction.

Classifiers	Classifiers with Autoencoder	ML without Dimensionality Reduction Technique
Logistic regression	0.875	0.865
Support vector machine	0.863	0.842
Random forest	0.828	0.814
Naïve Bayes	0.836	0.711

**Table 5 diagnostics-12-01796-t005:** Classifier with dimensionality reduction and feature selection technique.

Classifiers	Classifiers with Autoencoder	Classifiers with RFE
Logistic regression	0.875	0.840
Support vector machine	0.863	0.823
Random forest	0.828	0.822
Naïve Bayes	0.836	0.743

**Table 6 diagnostics-12-01796-t006:** Overall result analysis of proposed classifier.

Classifiers	Accuracy Score	Precision	Recall	F1 Score
Logistic regression	0.875	0.918	0.926	0.922
Support vector machine	0.863	0.867	0.971	0.916
Random forest	0.828	0.823	0.989	0.898
Naïve Bayes	0.836	0.893	0.901	0.897
1D-CNN	0.885	0.907	0.948	0.927

**Table 7 diagnostics-12-01796-t007:** Classifiers accuracy and F1 score analysis with various dimensionality reduction techniques.

ML Model	Accuracy			F1 Measure		
	PCA	LDA	Autoencoder	PCA	LDA	Autoencoder
Logistic regression	0.737	0.618	0.875	0.829	0.707	0.922
Support vector machine	0.842	0.625	0.863	0.910	0.716	0.916
Random forest	0.816	0.618	0.828	0.885	0.704	0.898
Naïve Bayes	0.770	0.625	0.836	0.856	0.714	0.897

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
