# Peer review of "An Unsupervised Neural Network Feature Selection and 1D Convolution Neural Network Classification for Screening of Parkinsonism"

_diagnostics, 2022, doi:10.3390/diagnostics12081796_

Round 1

Reviewer 1 Report

In this work, an effective data analysis method and meaningful research background are proposed, and the author presents a full data analysis process. Therefore, I suggest that the article be accepted after minor revision. Some suggestions are as follows:

(1) In the abstract, the full name of ML should be presented.

(2) In the introduction, CNN-1d appeared for the first time, and its full name should be given.

(3) The application overview of CNN should be presented. For example, CNN in target detection (eg, https://doi.org/10.1109/JSEN.2021.3103042 , signal identification (eg, https://doi.org/10.1109/JSEN.2022.3174251, image recognition (eg, https://doi.org/10.1016/j.knosys.2020.105887 , etc.

(4) In the conclusion, it will be beneficial for readers to explain by the items.

Author Response

Dear sir,

I am thankful and grateful for your comments and suggestions to improve the manuscript of my research paper.

Please note that I found your comments very useful, not only for this research paper but for my future work as well. I have tried my level best to implement all your suggestions/comments.

Below is a list of your comments (marked in Yellow) and my responses/implementation.

Should there be any further enhancement or improvement required to meet the standards of your esteemed journal please do not hesitate to contact me. 

Yours sincerely,

Anonymous

Comments and Suggestions for Authors

In this work, an effective data analysis method and meaningful research background are proposed, and the author presents a full data analysis process. Therefore, I suggest that the article be accepted after minor revision. Some suggestions are as follows:

  • In the abstract, the full name of ML should be presented. (Done)

Parkinson’s disease (PD) is the second most common neurodegenerative disorder after Alzheimer's disease. It has a slowly progressing neurodegenerative disorder rate. PD patients have multiple motor and non-motor symptoms, including vocal impairment. which is one of the main symptoms. The identification of PD based on vocal disorder is at the forefront of research. In this paper, an experimental study is performed on an open source Kaggle PD speech dataset and novel comparative techniques employed to identify PD. We proposed an unsupervised autoencoder feature selection technique and passed the compressed features to supervised machine learning (ML) algorithms. We also investigated the state-of-the-art deep learning 1D convolutional neural network (CNN-1D) for PD classification. In this study, the proposed algorithms are Support Vector Machine, Logistic Regression, Random Forest, Naive Bayes and CNN-1D. The classifier performance is evaluated in terms of accuracy score, precision, recall, and F1-score measure. The proposed 1D CNN model shows the highest result of 0.927% and Logistic Regression shows 0.922% on the benchmark dataset in terms of F1-measure. The major contribution of the proposed approach is that unsupervised neural network feature selection has not previously been investigated in Parkinson’s detection. Clinicians can use these techniques to analyse the symptoms presented by patients and based on the results of the above algorithms can diagnose the disease at an early stage, which will allow for improved future treatment and care.

  • In the introduction, CNN-1d appeared for the first time, and its full name should be given.(done).

In the proposed study, our aim is to use unsupervised autoencoder feature selection technique and then pass the optimal features to supervised ML models. We also use a 1D Convolutional Neural Network model to verify the deep learning solution for PD detection. The proposed approach, convolutional neural network-D (CNN-1D), has better results when compared with the autoencoder Feature selection technique.

  • The application overview of CNN should be presented. For example, CNN in target detection (eg, https://doi.org/10.1109/JSEN.2021.3103042 ), signal identification (eg, https://doi.org/10.1109/JSEN.2022.3174251), image recognition (eg, https://doi.org/10.1016/j.knosys.2020.105887 ), etc.  (Done)

The convolutional neural network (CNN) can extract feature automatically and brought revolution especially in the field of computer vision. The common applications of CNN in computer vision tasks are  [16] image classification [17], object detection [18], image segmentation [19], medical domain [20] agriculture domain [21] scene understanding [22] text classification.

(4) In the conclusion, it will be beneficial for readers to explain by the items.(done)

Initial stage of PD detection is a very important step in gaining a better understanding of the causes of the disease, thus ensuring the correct measures are implemented and therapeutic interventions are initiated. This study is an effort to outline a broad review of the diagnosis system of PD by applying various robust ML techniques. The different ML dimensionality reduction techniques are evaluated to achieve optimal features and make comparisons to obtain the best performance on the benchmark dataset. In this study, we performed analysis of PD disease in four different way. (a) evaluation of PD detection without any feature selection technique (b) analysis of four different machine learning based on three different dimensional reduction technique (c). We also perform analysis of recursive feature elimination technique with dimensionality reduction techniques. (4). In last section of our experimental approach, we also conduct our evaluation on 1D CNN with traditional machine learning models. We concluded that 1D-CNN model shows better results compared to the manual feature selection technique and 1D-CNN model shows the highest accuracy of 0.85% and Logistic Regression with autoencoder dimensionality reduction shows the highest accuracy of 0.75%. Based on this experimental evaluation, we can say that deep learning based 1D CNN shows better performance as compared to manual feature selection techniques-based machine learning models. The reason behind is that the deep learning model performs better with large number of features and have the capability of automatically feature selection. Our technique has some limitations that we do analysis on only one dataset and second factor that we do not consider about the dataset balancing methods.. In future studies, we are interested in exploring the ensemble and boosting techniques with different feature selection techniques on more than one PD dataset. The outcome of this work can be viewed as a promising first step towards the application of cutting-edge research for early detection of PD.

Reviewer 2 Report

This is an interesting manuscript using an unsupervised ANN for Feature Selection before passing data to other ML systems. The aim is to detect Parkin disease. The overall concept for this manuscript is very interesting and has important applications, however, there are mainly organizational and presentation issues that should be corrected. Follows a list of comments

There are numerous repetitions of the same concepts, for example in line 78 “In the proposed study, our aim is to use unsupervised…”, in line 97 “The scope of this study is the detection of PD at an early stage using the 1-D Convolutional Neural Network and traditional….” Line 104 “The main objective of this study is the….”, in my opinion the manuscript should be revised and be simplified thus improve readability….

Table 1, in the first column, is better to mention the first author name + et.al.

Liked figure 1 as shows the training/test procedure and helps the reader to understand the flow.

In line 92 is stated, “The hypothesis of this research study was that PD could be detected in its early stages in a person by changes in the characteristics of finger movement as they typed on a keyboard and that such changes could be used to distinguish and classify people with PD 94 from those without.” However, when reading the manuscript is found that the application is based on voice data to detect PD…. !!! please clarify or modify the manuscript accordingly.

Line 171 “The research scope of the proposed study, to the best of our knowledge….” A literature review is not the appropriate place to mention the pros of the current research, in my opinion, this paragraph could be among the last paragraphs of the discussion.

Figure 2 does not offer important information, in my opinion, is better to remove it and add the number of samples and relevant percentage for each group.

Figures 3 and 4 are not referred in the manuscript body, please add a reference for figure 3 and in my opinion figure 4 does not offer important information for SVM and could be removed. Similarly, all tables and figures should be referred in the manuscript by their numbers.

Section 4. Results and Discussion is not as usual !. Could the author separate the results in a different section and the discussion in a second section?

In Table 4. For classifiers with Dimensionality it is not clear what is presented (the OA?) could the author add also a statistical test comparing the performance of the approach with and without dimensionality reduction?  To support the superiority of the approach? A test for proportion would be OK. similarlly for the other tables showing performance.

Author Response

Dear sir,

I am thankful and grateful for your comments and suggestions to improve the manuscript of my research paper.

Please note that I found your comments very useful, not only for this research paper but for my future work as well. I have tried my level best to implement all your suggestions/comments.

In the attached document I have listed all your comments/suggestions (marked in Yellow)and my response/implementation. Furthermore, please note that you raised some organizational and presentation issues, although I have tackled all the concerns, however, there are some changes in the line numbers which I have mentioned in the document below.

Should there be any further enhancement or improvement required to meet the standards of your esteemed journal please do not hesitate to contact me. 

Yours sincerely,

Anonymous

Reviewer II

Comments and Suggestions for Authors

This is an interesting manuscript using an unsupervised ANN for Feature Selection before passing data to other ML systems. The aim is to detect Parkin disease. The overall concept for this manuscript is very interesting and has important applications, however, there are mainly organizational and presentation issues that should be corrected. Follows a list of comments

There are numerous repetitions of the same concepts, for example in line 78 “In the proposed study, our aim is to use unsupervised…”, in line 97 “The scope of this study is the detection of PD at an early stage using the 1-D Convolutional Neural Network and traditional….” Line 104 “The main objective of this study is the….”, in my opinion the manuscript should be revised and be simplified thus improve readability…. (done) (ne line no. 83, 224 &

Table 1, in the first column, is better to mention the first author name + et.al. (Done)

Liked figure 1 as shows the training/test procedure and helps the reader to understand the flow. (Done)

In line 92 is stated, “The hypothesis of this research study was that PD could be detected in its early stages in a person by changes in the characteristics of finger movement as they typed on a keyboard and that such changes could be used to distinguish and classify people with PD 94 from those without.” However, when reading the manuscript is found that the application is based on voice data to detect PD…. !!! please clarify or modify the manuscript accordingly. (Done)

Line 171 “The research scope of the proposed study, to the best of our knowledge….” A literature review is not the appropriate place to mention the pros of the current research, in my opinion, this paragraph could be among the last paragraphs of the discussion. (Done)

Figure 2 does not offer important information, in my opinion, is better to remove it and add the number of samples and relevant percentage for each group. (Done)

Figures 3 and 4 are not referred in the manuscript body, please add a reference for figure 3 and in my opinion figure 4 does not offer important information for SVM and could be removed. Similarly, all tables and figures should be referred in the manuscript by their numbers. (Done)

Section 4. Results and Discussion is not as usual !. Could the author separate the results in a different section and the discussion in a second section? (Done)

In Table 4. For classifiers with Dimensionality it is not clear what is presented (the OA?) could the author add also a statistical test comparing the performance of the approach with and without dimensionality reduction?  To support the superiority of the approach? A test for proportion would be OK. similarlly for the other tables showing performance.

I would like to mention that our approach is the analysis of different machine learning classifiers based on pd speech dataset by using three different types of dimensionality reduction techniques. We are also making comparison with deep learning classifier 1D CNN  with the supervised machine learning classifiers . We find that 1D CNN classifier has the better performance as compared to ML classifiers with dimensionality reduction and without dimensionality reduction techniques. 

Reviewer 3 Report

The authors used the kaggle dataset from the KAGGLE. In fact, too many papers handled this dataset and the performance has been matured. The additional techniques seem to be overfitting.

The application of autoencoder in this research field has already been reported. For example, "End-to-end deep learning approach for Parkinson’s disease detection from speech signals, Biocybernetics and Biomedical Engineering, 2022"

PCA was also used in several previous studies. "Performance Evaluation of Combined Feature Selection and Classification Methods in Diagnosing Parkinson Disease Based on Voice Feature"

The feature selection techniques are not new. PCA and LDA are not interesting. Since the dataset is relatively small, I don't know whether the feature selection is effective in this form. 

The authors did not show how to handle the voice dataset. There are several ways to analyze the sound with algorithms and signal processings. 

Did the authors perform data augmentation? How about a noise reduction process?

The 10-fold cross validation is necessary to confirm the performance.

Limitation - no clinical validation.

In summary, the authors should highlight the strengths of their study compared with the previous literature. I cannot find the academic or clinical merit of the current form. 

Author Response

Dear sir,

I am thankful and grateful for your comments ans suggestions to improve the manuscript of my research paper.

Please note that I found your comments very useful, not only for this research paper but for my future work as well. I have tried my level best to implement all your suggestions/comments.

The attached file contains your comments (marked in Yellow) and my responses/implementation.

Should there be any further enhancement or improvement required to meet the standards of your esteemed journal please do not hesitate to contact me. 

Yours sincerely,

Anonymous

Reviewer III

Comments and Suggestions for Authors

The authors used the Kaggle dataset from the KAGGLE. In fact, too many papers handled this dataset and the performance has been matured. The additional techniques seem to be overfitting.

The application of autoencoder in this research field has already been reported. For example, "End-to-end deep learning approach for Parkinson’s disease detection from speech signals, Biocybernetics and Biomedical Engineering, 2022"

PCA was also used in several previous studies. "Performance Evaluation of Combined Feature Selection and Classification Methods in Diagnosing Parkinson Disease Based on Voice Feature"

The feature selection techniques are not new. PCA and LDA are not interesting. Since the dataset is relatively small, I don't know whether the feature selection is effective in this form. 

The authors did not show how to handle the voice dataset. There are several ways to analyze sound with algorithms and signal processing. 

Did the authors perform data augmentation? How about a noise reduction process?

The 10-fold cross-validation is necessary to confirm the performance.

Limitation - no clinical validation.

In summary, the authors should highlight the strengths of their study compared with the previous literature. I cannot find the academic or clinical merit of the current form. 

As this dataset is openly available and health dataset has privacy, therefore, finding the real dataset is not an easy task for research purpose. To avoid this issue, we used the Kaggle public source dataset. No doubt a lot of research work has been done in this field using the same dataset, however, in this study, we were not only focused on simple implementation and analysis, we made the comparative analysis of three different dimensionality reduction techniques with each other and without Machine Learning classifier.  The different researchers have used the principal component analysis (PCA) and Autoencoder (AE) for dimensionality reduction in different state of the art technique, however, these were individually analysed and have not made any comparison with the 1D light weight Convolutional Neural Network. 

As regarding the problem of dataset, dataset is not small, it contains 576 features, these are too large in machine learning context, so there is need of proper feature selection. Beside that, I would like to mention in this study we used the random oversampling technique to balance the dataset. As the dataset is not imaging, so our proposed approach in term of augmentation is random oversampling.  We also used the 10-fold cross validation to measure the performance of four proposed machine learning classifier.

In summary, the focus of this paper is the analysis of machine learning classifiers by using the different dimensionality reduction techniques and make the comparison with the state of art 1D CNN classifier. In this study, we evaluated that the deep learning approaches is better than the manual feature selection technique of machine learning model for the analysis and classification of Parkinson speech dataset analysis.

Round 2

Reviewer 2 Report

The authors have responded successfully to the review comments, in my opinion, the manuscript is worth appearing in Diagnostics.

Reviewer 3 Report

"The outcome of this work can be viewed as a promising first step towards the application of cutting-edge research for early detection of PD."

- I cannot agree with this comment. Considering the recent field of speech recognition, I think the level of technology presented in this study is low.

I won't give more suggestions for revision because the editor and reviewers are positive about publishing.  I express my negative intention to publish because my previous suggestions were not accepted by the author. The author didn't even discuss the research I mentioned.